# Integration of an OS-Based Machine Learning Score (AS Score) and Immunoscore as Ancillary Tools for Predicting Immunotherapy Response in Sarcomas

**DOI:** 10.3390/cancers17152551

**Published:** 2025-08-01

**Authors:** Isidro Machado, Raquel López-Reig, Eduardo Giner, Antonio Fernández-Serra, Celia Requena, Beatriz Llombart, Francisco Giner, Julia Cruz, Victor Traves, Javier Lavernia, Antonio Llombart-Bosch, José Antonio López Guerrero

**Affiliations:** 1Pathology Department, Instituto Valenciano de Oncología, Calle Gregorio Gea 31, 4to Piso, 46009 Valencia, Spain; julia.cruz@uv.es (J.C.); vtraves@fivo.org (V.T.); 2Patologika Laboratory, Hospital QuirónSalud, 46010 Valencia, Spain; 3Pathology Department, University of Valencia, 46010 Valencia, Spain; francisco.giner@uv.es (F.G.); antonio.llombart@uv.es (A.L.-B.); 4Centro de Investigación Biomédica en Red de Cáncer (CIBERONC), 28029 Madrid, Spain; 5Molecular Biology Department, Instituto Valenciano de Oncología, 46009 Valencia, Spain; rlopez@fivo.org (R.L.-R.); afernandez@fivo.org (A.F.-S.); 6Joint Cancer Research Unit, IVO-CIPF, 46009 Valencia, Spain; jalopez@fivo.org; 7Pathology Department, Hospital Universitario y Politécnico “La Fe”, 46009 Valencia, Spain; giner_edu@gva.es; 8Dermatology Department, Instituto Valenciano de Oncología, 46009 Valencia, Spain; crequena@fivo.org (C.R.); bllombart@fivo.org (B.L.); 9Oncology Department, Instituto Valenciano de Oncología, 46009 Valencia, Spain; javilavernia@gmail.com; 10Pathology Department, Catholic University of Valencia, 46001 Valencia, Spain

**Keywords:** angiosarcomas, cutaneous angiosarcomas, non-cutaneous angiosarcomas, AS score, Immunoscore, sarcoma prognosis

## Abstract

Angiosarcomas (ASs) are a highly aggressive subset of sarcomas. This study aimed to develop and validate an immune-related prognostic model, termed the AS score, using two independent sarcoma cohorts. The model was initially established using a previously characterized cohort of 25 ASs. Validation was performed using transcriptomic data from TCGA Sarcoma cohort. The AS score effectively stratified AS cases into two prognostically distinct groups. Cases with a high AS score exhibited significantly poorer outcomes. Validation in TCGA sarcoma cohort confirmed the prognostic utility of both the AS score and the Immunoscore. Further survival analyses integrated these scores into four categories. Notably, among tumors with a high Immunoscore, the AS score significantly distinguished outcomes (*p* < 0.0001), identifying a subset with poor prognosis but potential sensitivity to immunotherapy. The combined use of the AS score and Immunoscore provides significant prognostic value in sarcoma, indicating that patients with high scores in both metrics may be suitable candidates for immunotherapy.

## 1. Introduction

Sarcomas are a diverse and heterogeneous group of malignant tumors of mesenchymal origin, collectively accounting for approximately 1% of adult solid tumors [1,2]. With over 70 histological subtypes identified, they exhibit a wide spectrum of clinical behaviors and biological characteristics, many of which are highly aggressive and resistant to systemic therapy [1,2]. Among them, angiosarcomas (ASs) represent one of the most aggressive and heterogeneous subtypes, often associated with poor progression-free survival (PFS) and poor overall survival (OS) [3,4,5,6,7,8,9,10,11,12,13,14,15,16,17]. ASs represent between 1 and 2% of all soft tissue sarcomas and are typically found in older individuals (60–80 years old) [1,2,3,4,5,6,7,8,9,10,11,12,13,14,15,16,17]. The most common location (50–60%) is the skin/subcutaneous tissue, with more than half occurring on the scalp and face [1,2,3,4,5,6,7,8,9,10,11,12,13,14,15,16,17]. Mammary cases are much less frequent (5–10%), usually following radiotherapy for breast cancer [2,9,13]. Visceral cases (<5%), such as hepatic or splenic, are rare [2,3,4,5,7]. ASs associated with chronic lymphedema (Stewart–Treves syndrome) are also uncommon [2,3,5,8,9,12].

ASs are broadly categorized into primary and secondary forms [2,4,5,6,7,8,9,10,11,12,13,14,15,16,17,18,19,20]. The secondary form often arises in the context of prior radiotherapy or chronic lymphedema and is characterized by distinct molecular profiles, such as *MYC* amplification [2,4,5,6,7,8,9,10,11,12,13,14,15,16]. Despite advances in cytotoxic therapies and surgical approaches [21,22,23,24], advanced/metastatic ASs remain largely refractory to most conventional treatments, underscoring the urgent need for alternative therapeutic strategies—particularly in metastatic/recurrent AS as well as in unresectable AS.

Immunotherapy, particularly immune checkpoint inhibitors (ICIs), has emerged as a promising therapeutic approach for several malignancies that have previously been resistant to treatment [25,26,27,28,29,30,31,32,33,34]. While ICIs, either alone or in combination with polychemotherapy, are currently approved for certain tumors in the neoadjuvant setting and as first-line systemic therapy for advanced or metastatic disease, their role in angiosarcomas and other sarcoma subtypes is not yet fully understood [25,26,27,28,29,30,31,32,33,34]. Recent studies suggest that a subset of ASs—especially those located in the head and neck region or lacking *MYC* amplification—may possess a more immunogenic tumor microenvironment (TME), potentially making them more responsive to immunotherapy [10,12,16,17,19,20,27,28]. In this context, the integration of immune-related biomarkers, such as the Immunoscore, has been proposed as a method to assess the immunologic activity of the TME [10,12,16,17,19,20].

Building upon our previous work characterizing the immune microenvironment in ASs through transcriptomic profiling and immunohistochemistry [12], we have developed and validated an immune-related prognostic model—the AS score—based on the expression of selected immune-associated genes. The validation step of the prognostic model was performed on the soft tissue sarcoma cohort from The Cancer Genome Atlas (TCGA) [35]. Additionally, we explored the integration of the AS score with the Immunoscore to refine risk stratification and evaluate their potential as ancillary tools for identifying patients who may benefit from immunotherapy in the context of sarcomas.

## 2. Material and Methods

### 2.1. Data Acquisition and Cohort Description

#### 2.1.1. Training Cohort/Series

This study was designed as a validation analysis of an immune-related prognostic model previously developed using transcriptomic data from a series of 25 formalin-fixed, paraffin-embedded (FFPE) angiosarcoma (AS) tissue samples (Table 1) [12].

Formalin-fixed, paraffin-embedded (FFPE) tissue samples, encompassing both cutaneous and non-cutaneous angiosarcomas (including soft tissue and visceral forms; see Figure 1), were collected from patients diagnosed between 2000 and 2015 and retrieved from the pathology archives of multiple hospitals in our region. Patient age, sex, tumor size, medical history, anatomical location, treatment modalities (surgery, chemotherapy, radiotherapy), type of surgical procedure, and clinical outcomes (alive or deceased due to disease progression) were abstracted from medical records.

This study received approval from the Institutional Review Board and was conducted in accordance with the ethical guidelines of the participating institution and the principles outlined in the Declaration of Helsinki (1975, as amended in 2008). This cohort was utilized to develop the prognostic model, referred to as the “AS score”.

#### 2.1.2. Validation Cohort/Series

To assess the prognostic applicability of the developed “AS score” in a broader sarcoma context, the model was applied to the soft tissue sarcoma cohort from The Cancer Genome Atlas (TCGA), which includes 253 adult cases (Appendix A) comprising dedifferentiated liposarcoma, leiomyosarcoma, undifferentiated pleomorphic sarcoma, myxofibrosarcoma, malignant peripheral nerve sheath tumor, and synovial sarcoma [35].

Given the absence of a dedicated angiosarcoma series within TCGA, the entire sarcoma cohort was used as a surrogate validation set.

Transcriptomic and clinical data were accessed through the firebrowse platform (http://firebrowse.org/) publicly available matrices.

### 2.2. Immuno-Related Gene Expression Data (HTG EdgeSeq Precision Immuno-Oncology Assay)

Gene expression was quantified using the HTG EdgeSeq Precision Immuno-Oncology Assay (HTG Molecular Diagnostics, Tucson, AZ, USA) [12], which profiles the immune contexture within tumor microenvironments. This targeted panel measures the expression of 1392 immune-related mRNAs across multiple tumor types. Expression values were normalized both within and between samples: we applied median-of-ratios normalization via DESeq2 (v1.42.1) [36] to reduce technical variability and emphasize biologically meaningful differences.

### 2.3. Development and Validation of Prognostic AS Score

The AS score was based on the expression data of candidates’ immune-related genes from 25 AS cases [12]. A multi-step analytical approach was employed to build the model. Initially, the prognostic significance of each of the 1392 genes under consideration was assessed using the Maxstat algorithm (R package Maxstat, v0.7-25), which determines the optimal cut-off based on log-rank test statistics for overall survival, thereby allowing the identification of genes with the strongest univariate associations with patient outcomes. Following the screening, the ten genes showing the highest statistical significance, as indicated by the lowest *p*-values, were selected for further analysis (Appendix A). These ten top-ranked candidates were subsequently entered into a multivariate Cox regression to evaluate their independent prognostic value and assign their weights (Figure 2). As a result of this analysis, four genes—IGF1R, MAP2K1, SERPINE1, and TCF12—were identified (Appendix A). Based on these four genes, the prognostic risk score, referred to as the AS score, was developed. The AS score was defined as the sum of the normalized expression values of the four genes, each weighted by its corresponding regression coefficient derived from the multivariate Cox analysisAS score = (IGF1R × −1.22281) + (MAP2K1 × −0.63790) + (SERPINE1 × 1.00977) + (TCF12 × 0.61161)

A cut-off value of −1.9525 was established based on the value that best stratified the population according to OS. This value was used to dichotomize patients into high- and low-risk groups.

The developed model was applied to TCGA sarcoma cohort re-establishing the cut-off value to assess its prognostic value in an independent population. The new cut-off was 0.85. Gene expression data was retrieved as read counts and median normalized with DESeq2 [36] to compare this data with our series. Clinical metadata included histological subtype, tumor location, patient sex, and overall survival information. The AS score was calculated for all TCGA samples. To assess immune cell infiltration, the ESTIMATE R package (v1.0.13) [37] was used. Immunoscore values were normalized and dichotomized into high and low groups based on the most significative cut-off value regarding OS by the Maxstat algorithm.

### 2.4. Statistical Analysis

Nonparametric tests, including the Wilcoxon rank-sum and Kruskal–Wallis tests, were used to analyze continuous variables. Survival outcomes were assessed using Kaplan–Meier estimates, with group differences evaluated via the log-rank test. Multivariate analyses were performed with Cox proportional-hazards regression models. A two-sided *p*-value < 0.05 was considered statistically significant in all analyses. The primary time-to-event endpoint was overall survival (OS), defined as the time from diagnosis to death. All statistical analyses were conducted in RStudio version 4.3.3.

## 3. Results

### 3.1. Development of the AS Score in the AS Series

The total number of interrogated immune-related genes was categorized using the optimal cutoff value determined by the Maxstat algorithm. Genes were then selected using the log-rank test, applying a stringent *p*-value threshold of 0.002 to reduce the number of features used as input for the Cox regression model. The normalized expression levels of the 10 most significant genes were used to train the Cox regression model. An overall survival (OS) score was calculated using the coefficients of the significant parameters and their normalized expression levels, as followsScore = IGF1R* × −1.22281 + MAP2K1* × −0.63790 + SERPINE1* × 1.00977 + TCF12* × 0.61161

Samples with high scores, including both cutaneous and non-cutaneous localizations, exhibited significantly poorer prognosis (*p* = 0.00012) compared to low-score samples, which were predominantly cutaneous (Figure 3).

To further validate the prognostic value of the AS score, we conducted both univariate and multivariate Cox regression analyses, incorporating common clinicopathological parameters alongside the AS score (Appendix A). In the univariate analysis, the primary tumor site (cutaneous vs. non-cutaneous; *p* = 0.0012), surgical margin status (*p* = 0.0049), and chemotherapy treatment (*p* = 0.0003) were significantly associated with overall survival (OS). However, in the multivariate model, only the AS score remained an independent predictor of poor prognosis (HR = 7.0; 95% CI: 2.4–21; *p* = 0.000444), while the other clinical variables lost statistical significance. These findings support the robustness of the AS score as a prognostic biomarker that provides independent predictive value beyond conventional clinicopathological features.

### 3.2. Validation of the AS Score in TCGA Sarcoma Cohort

The OS-based immune prognostic score (AS score) was applied to TCGA sarcoma dataset, which includes 253 soft tissue sarcoma cases. Patients were stratified into high- and low-risk groups based on a pre-established cut-off value of −1.9525. The AS score showed a significant correlation with overall survival, effectively distinguishing two prognostic groups with distinct clinical outcomes (*p* = 0.0006; Figure 4A). Patients in the high-risk group had shorter survival times compared to those in the low-risk group. These findings replicate the results previously observed in the angiosarcoma cohort, further validating the AS score as a robust and independent prognostic tool across a broader sarcoma population.

### 3.3. Prognostic Relevance of the Immunoscore

The Immunoscore, calculated using the ESTIMATE algorithm, was also evaluated in TCGA sarcoma cohort. Patients with a high Immunoscore demonstrated significantly improved overall survival (*p* = 0.0029), supporting the role of immune infiltration as a favorable prognostic factor. Notably, a positive correlation was observed between the Immunoscore and the AS score when analyzed as continuous variables (*p* = 2.9 × 10^−8^), suggesting that tumors with greater immune infiltration also tended to exhibit higher AS scores (Figure 4B). These findings indicate that immune-hot, yet high-risk tumors may constitute a distinct subgroup characterized by both aggressive behavior and potentially actionable immunogenicity.

### 3.4. Combined Stratification by AS Score and Immunoscore

To further investigate the prognostic landscape, patients were stratified based on their AS score and Immunoscore. Prognostic associations with overall survival were assessed independently for each score, as well as in combination. Patients were categorized into four groups based on their AS score and Immunoscore status: Group 1: High AS score/High Immunoscore; Group 2: High AS score/Low Immunoscore; Group 3: Low AS score/High Immunoscore and Group 4: Low AS score/Low Immunoscore.

The combination of both classifiers demonstrated significant prognostic stratification (*p* = 0.00021; Figure 5A). Notably, within the low Immunoscore cohort, the AS score did not significantly stratify patients. In contrast, among tumors with a high Immunoscore, the AS score showed strong prognostic power (*p* < 0.0001; Figure 5B). These findings highlight the potential utility of a combined immune-risk model to enhance patient selection in future immunotherapy-based clinical trials, with patients in Group 1 being eligible candidates for immunotherapy.

Furthermore, multivariate Cox regression analyses—including both the AS score and Immunoscore as covariates—confirmed that each provides independent prognostic information. This supports the hypothesis that the AS score and Immunoscore reflect distinct, yet potentially complementary, biological processes relevant to tumor progression and immune contexture (Figure 6).

To rigorously assess whether the AS score offers prognostic value independent of established clinicopathological variables, we conducted an extended multivariate Cox regression analysis in TCGA sarcoma cohort, adjusting for age, gender, tumor size, mitotic rate, radiotherapy, and surgical margin status (Appendix A). Significantly, both the AS score (HR = 2.47; 95% CI: 1.42–4.30; *p* = 0.0014) and the Immunoscore (HR = 0.42; 95% CI: 0.25–0.70; *p* = 0.0009) retained statistical significance in this comprehensive model, whereas none of the clinical variables, including tumor size, mitotic rate, or surgical margin status did. These findings confirm that the AS score is an independent prognostic marker, offering robust, biology-driven prognostic information that complements traditional clinical features in soft tissue sarcomas.

## 4. Discussion

In a recently published study, we reported a series of cutaneous and non-cutaneous angiosarcomas, which were stratified into tumors with a ‘hot’ immune microenvironment and those with a weaker immune response [12]. Hypothetically, these ‘hot’ tumors could benefit to some extent from immunotherapy, although other genetic, histological, and clinical factors may also influence the response to such treatment [24,25,26,27,28,29,30,31,32,33,34]. While some angiosarcomas may be susceptible to immunotherapy, to the best of our knowledge, no studies have stratified these neoplasms using scores based on immunity-associated genes and their correlation with overall survival. Furthermore, the use of immune microenvironment-related scores linked to overall survival has not been extensively explored in other types of sarcomas.

An AS score, based on the expression of four immune-related genes, was defined in the clinical context of angiosarcoma (AS) in the present study. Score-based stratification identifies a subgroup of cutaneous ASs with poorer prognosis, which correlates with a higher Immunoscore. Our findings align with previous observations in angiosarcomas, where tumors exhibiting greater immune activity—i.e., a ‘hot’ microenvironment—particularly those lacking *MYC* amplification or located in the head and neck region, were associated with more favorable outcomes [10,12,16,17,19,20,27,28]. In this study, we extend these insights to a heterogeneous cohort of soft tissue sarcomas [35], highlighting that immune-related transcriptomic features may have broad prognostic significance across different histologic subtypes. The robustness of the AS score in this independent cohort further supports its predictive value beyond vascular sarcomas.

In addition, we independently validated the developed immune-related prognostic model (AS score), within a broader soft tissue sarcoma cohort from The Cancer Genome Atlas (TCGA) [36]. The model allows the successful stratification of patients into high- and low-risk groups, demonstrating statistically significant differences in overall survival. These findings reinforce the prognostic value of the AS score and support its potential applicability beyond angiosarcoma. However, future studies should validate each score separately across diverse histological subtypes and consider recalibrating classification thresholds to account for tumor-specific biological contexts.

In parallel, we evaluated the Immunoscore derived from gene expression signatures indicative of immune cell infiltration. A higher Immunoscore was significantly associated with improved survival, reinforcing the concept that the tumor immune microenvironment plays a critical role in modulating sarcoma progression [25,26,27,28,29,30,31,32,33,34]. Interestingly, we observed a significant positive correlation between the Immunoscore and the AS score, suggesting that some tumors exhibit high immunogenicity despite being classified as high-risk by the prognostic model. These “immune-hot and high-risk” tumors may represent a biologically distinct subset characterized by both aggressive behavior and potential sensitivity to immunomodulatory therapies.

To further refine prognostication, we combined the AS score and the Immunoscore, generating four risk groups based on both immune activity and a transcriptomic-based AS score. This integrated approach improved survival stratification and identified specific subgroups with markedly different outcomes. Notably, the prognostic impact of the AS score was preserved in tumors with high immune infiltration, while it was diminished in immune-cold tumors. These results suggest that combining immune-related transcriptomic features with estimates of immune infiltration may enhance risk prediction and guide patient selection for future immunotherapy-based strategies.

This study has some limitations. The retrospective nature of the analysis, combined with the lack of a dedicated angiosarcoma cohort in TCGA, limits the specificity of the validation. Furthermore, the AS score was applied without cohort-specific recalibration, although supporting robustness may impact performance in histologically diverse sarcomas. In addition, we used a conventional statistical method, even though there are various advanced deep learning techniques [38,39,40] that could provide more detailed insights into the underlying immune mechanisms of certain biological processes and complement our research. Given the potential value of applying deep learning models, it would be interesting to employ them in future studies to validate our findings using prospective, independent angiosarcoma cohorts and other sarcoma types. Further clinical correlations are needed to confirm the model’s utility in guiding therapeutic decisions.

## 5. Conclusions

In conclusion, our findings validate the AS score as a prognostic tool for soft tissue sarcomas and demonstrate that its combination with the Immunoscore refines risk stratification. The combination of the AS score, based on the expression of four immune-related genes, together with Immunoscore-based classification presented a prognostic impact in sarcoma patients. Due to its hypothesized hot microenvironment (high Immunoscore) and worse prognosis (high AS score), these findings could constitute a window of opportunity for immunotherapy in this subgroup. These scores could constitute a valuable ancillary tool as selection criteria for clinical trials based on immunotherapy, particularly in the context of AS. However, the results should be validated in a wider series of angiosarcomas.

## Figures and Tables

**Figure 1 cancers-17-02551-f001:**
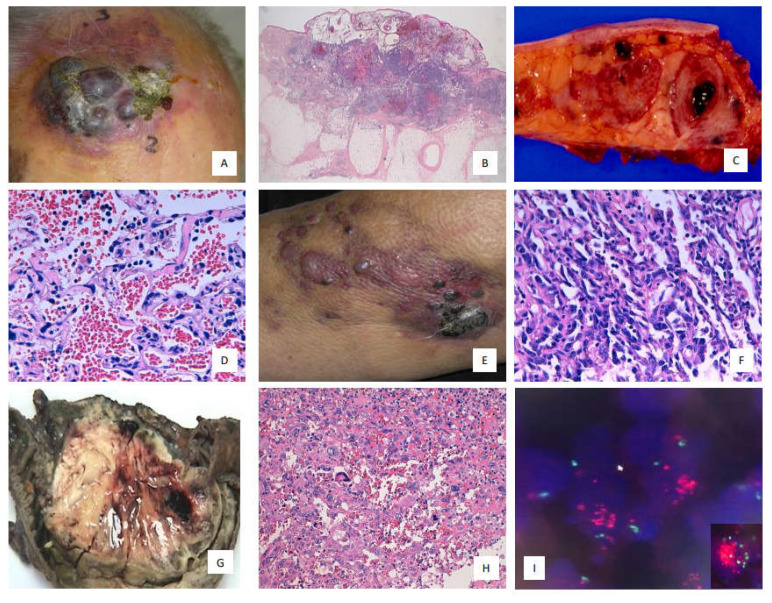
(**A**) Cutaneous angiosarcoma of the scalp; (**B**) Primary cutaneous angiosarcoma with prominent lymphoid infiltration (hematoxylin and eosin [H&E], 20×); (**C**) Primary breast angiosarcoma with hemorrhagic areas; (**D**) Primary breast angiosarcoma showing extensive vascular formation, endothelial atypia, and sparse lymphoid infiltration (H&E, 40×); (**E**) Lymphedema-associated cutaneous angiosarcoma; (**F**) Cutaneous angiosarcoma with spindle cell proliferation and minimal lymphoid infiltration (H&E, 40×); (**G**) Primary intestinal angiosarcoma with a fleshy appearance and hemorrhagic areas; (**H**) Primary intestinal angiosarcoma with pleomorphic morphology, mitotic figures, and marked nuclear atypia (H&E, 40×); (**I**) *MYC* amplification in secondary cutaneous angiosarcoma, 100×.

**Figure 2 cancers-17-02551-f002:**
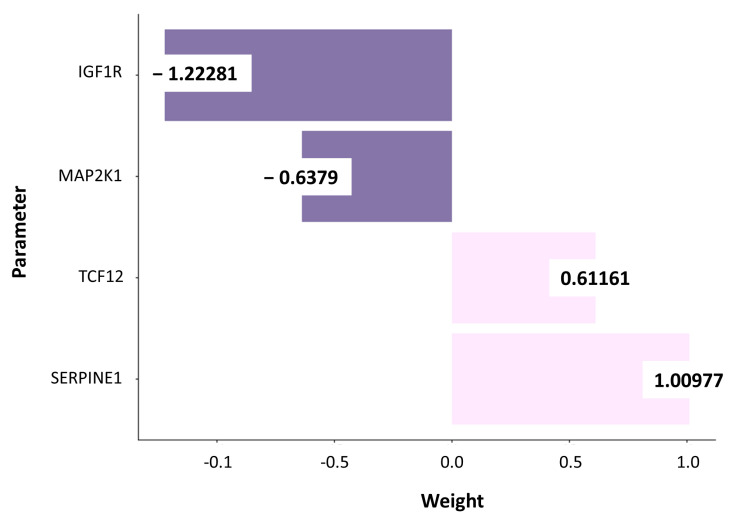
Immuno-based AS score performance. Model weights from selected parameters.

**Figure 3 cancers-17-02551-f003:**
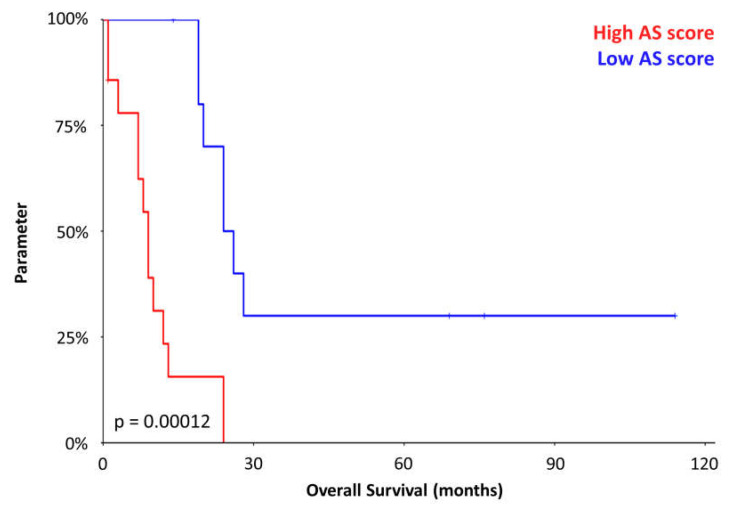
Immuno-based performance. The clinical impact of model-based stratification.

**Figure 4 cancers-17-02551-f004:**
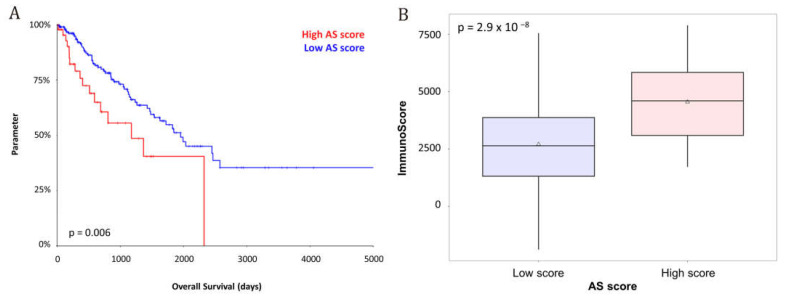
(**A**) Prognostic impact on the overall survival of the AS score; (**B**) Distribution of the Immunoscore between AS score-based groups.

**Figure 5 cancers-17-02551-f005:**
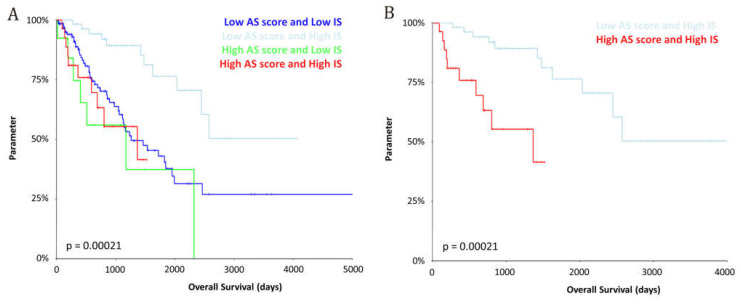
Impact of the AS score and Immunoscore combined classification by a long-rank test regarding: (**A**) four prognostic groups and (**B**) high Immunoscore groups.

**Figure 6 cancers-17-02551-f006:**
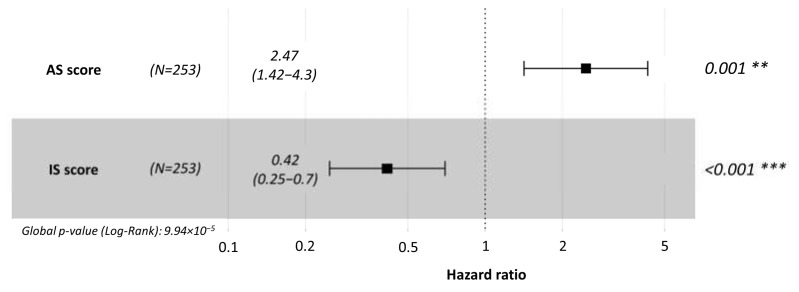
Multivariate Cox regression analysis demonstrating the independent prognostic value of the AS score and the Immunoscore. ** *p*-value < 0.01 and *** *p*-value < 0.001.

**Table 1 cancers-17-02551-t001:** Main clinicopathological characteristics of the cohort (*n* = 25).

Patient Characteristics		Tumor Characteristics	
**Age at diagnosis (years)**		**Tumor size (cm)**	
Median	71	Median	8
Range	31–83	Range	2–50
**Gender**		**Mitotic rate**	
Male	10 (40)	Median	20
Female	15 (60)	Range	0.0–37.0
**Primary Site**		**Margins**	
Cutaneous	17 (68)	Negative	21 (84)
non-cutaneous soft tissue	2 (8)	Positive	4 (16)
non-cutaneous visceral	6 (24)	**Surgery**	
**Follow-Up and Outcomes**		Yes	22 (88)
**Overall survival**		No	3 (12)
N	25	**Chemotherapy treatment**	
Events	20	Yes	9 (36)
Median (range)	14 (1–114) months	No	16 (64)
		**Radiotherapy treatment**	
		Yes	2 (8)
		No	23 (92)
		**MYC (FISH)**	
		Non-amplified	10 (40)
		Amplified	7 (28)
		NP	8 (32)

## Data Availability

The datasets generated and/or analyzed during this study are publicly available upon reasonable request.

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
