# Peer review of "Integration of an OS-Based Machine Learning Score (AS Score) and Immunoscore as Ancillary Tools for Predicting Immunotherapy Response in Sarcomas"

_cancers, 2025, doi:10.3390/cancers17152551_

Round 1

Reviewer 1 Report

Comments and Suggestions for Authors

See attached file

Author Response

-Reviewer 1:

We hereby submit our revised manuscript to be reconsidered for publication in Cancers following a revision in accordance with the reviewers’ comments.

We would like to thank the reviewers for the evaluation of our manuscript and for their constructive suggestions which we believe have contributed towards improving the article.

Please find below our response to the comments. The adapted sections of the article are highlighted in yellow in the manuscript.

Response: Thank you for this comment, with which we fully agree. It’s true that we used a conventional statistical method, even though there are various advanced deep learning techniques that could provide more detailed insights into the underlying immune mechanisms of certain biological processes and complement our research. We have added a note in the Discussion acknowledging that, given the potential value of applying deep learning models, it would be interesting to employ them in future studies to validate our findings using prospective, independent angiosarcoma cohorts and other sarcoma types.

Reviewer 2 Report

Comments and Suggestions for Authors

The article “Integration of an OS-Based Machine Learning Score (AS-Score) and Immunoscore as Ancillary Tools for Predicting Immunotherapy Response in Sarcomas” aims to create and validate an immune-related prognostic model (AS-Score) in two independent sarcoma cohorts:
- Was the sample size (n=25 in the training cohort) sufficient to create a generalizable prognostic model for angiosarcoma?
- How might the lack of angiosarcoma-specific data in the TCGA cohort affect the external validity of the AS score?
- Were the gene selection thresholds (e.g. p < 0.002) appropriate to avoid overfitting while maintaining biological significance?
- Was dichotomization of the immunoscore using the median a robust choice for all sarcoma subtypes?
- Does the AS score provide practical prognostic value beyond existing clinicopathologic parameters?
- Can the four-gene model (IGF1R, MAP2K1, SERPINE1, TCF12) be translated into a practical clinical test?
- What are the clinical implications of identifying tumors with a high immunoscore and a high ASscore - are they suitable for therapeutic intervention?
- What is the mechanistic rationale behind the selection of genes for AS score - do they play a known role in immune evasion or tumor progression in sarcomas?
- Is the observed correlation between AS score and immunoscore indicative of causality, or could it reflect overlapping but independent immune activity?
- How should the limitations of the model, particularly its retrospective design and single-region sample collection, affect its clinical application?
- What ethical safeguards should be put in place before using the AS score as an inclusion criterion for clinical trials?
- Could the AS score and immunoscore be adapted or retrained for other aggressive sarcoma subtypes?

Author Response

Please see the PDF attached

-Reviewer 2:

Comments and Suggestions for Authors

We hereby submit our revised manuscript to be reconsidered for publication in Cancers following a revision in accordance with the reviewers’ comments.

We would like to thank the reviewers for the evaluation of our manuscript and for their constructive suggestions which we believe have contributed towards improving the article.

Please find below our response to the comments. The adapted sections of the article are highlighted in yellow in the manuscript.

The article “Integration of an OS-Based Machine Learning Score (AS-Score) and Immunoscore as Ancillary Tools for Predicting Immunotherapy Response in Sarcomas” aims to create and validate an immune-related prognostic model (AS-Score) in two independent sarcoma cohorts:

  • Was the sample size (n=25 in the training cohort) sufficient to create a generalizable prognostic model for angiosarcoma?

Response: We acknowledge that a sample size of 25 is insufficient to develop a fully generalizable prognostic model. However, given the rarity of angiosarcoma, in which few large cohorts exist, this sample size represents a reasonable proof-of-concept starting point. As emphasized throughout the manuscript, this study serves to demonstrate feasibility and generate preliminary insights; it will require validation and refinement in larger, independent cohorts to achieve broader applicability.

  • How might the lack of angiosarcoma-specific data in the TCGA cohort affect the external validity of the AS score?

Response: Ideally, validation using an independent  large cohort comprised exclusively of angiosarcoma cases would provide the most reliable results. However, such datasets are currently unavailable, both internally and in public repositories. As an alternative, we evaluated the AS-score within the TCGA pan-sarcoma cohort to determine whether the model retains prognostic relevance in a broader, heterogeneous sarcoma population. While this approach is suboptimal, it offers initial insights into the score’s robustness across various sarcoma types.

  • Were the gene selection thresholds (e.g. p < 0.002) appropriate to avoid overfitting while maintaining biological significance?

   Response: A p‑value threshold of 0.002 was applied exclusively during the Maxstat filtering     step to reduce the number of genes considered for subsequent modeling. This threshold represents an optimal balance between statistical rigor and retaining a sufficient gene set for adjustment in the Cox regression analysis. It was used solely to identify the most prognostically informative genes for inclusion in the multivariate Cox regression—not to train or validate the model directly. Consequently, the risk of overfitting is more pertinent to the model‐training phase than to this preliminary selection step.

  • Was dichotomization of the immunoscore using the median a robust choice for all sarcoma subtypes?

Response: We appreciate this observation. In our study, we did not dichotomize the Immunoscore at the median. Instead, a data-driven method—the Maxstat algorithm—was used to determine the optimal cutoff for survival stratification. This approach ensures the threshold chosen offers maximal prognostic separation rather than relying on an arbitrary split . We acknowledge the discrepancy and have corrected this statement in the revised manuscript to accurately reflect our methodology. Moreover, we agree that any score intended for clinical application must be calibrated specifically to the sarcoma subtype and therapeutic context.

o Section 2.3. Development and Validation of Prognostic AS-Score, line 187; “Immunoscore values were normalized and dichotomized into high and low groups based on the most significative cut-off value regarding OS by maxstat algorithm.”

  • Does the AS score provide practical prognostic value beyond existing clinicopathologic parameters?

Response: We acknowledge the importance of assessing the added value of the AS‑score relative to standard clinicopathological variables. Although our current study focuses primarily on molecular features, we included additional multivariate analyses that integrate clinical variables. In both, the AS cohort and TCGA cohort, only AS score and IS score retain statistical significance in the multivariate analysis, while demonstrating their independent prognostic role. Regarding clinicopathological paraemeters, univariate analysis resulted significative for margins and location in the caso of AS cohor and margins and mitotic rate in the case of TCGA cohort. Nonetheless, the AS‑score may offer a complementary, biology-informed perspective that enhances existing stratification methods.

Section 3.3. Prognostic Relevance of the Immunoscore  , Line 264   “To further assess the clinical relevance of the AS-score, we performed multivariate Cox regression analyses that included standard clinicopathologic parameters alongside both the AS-score and Immunoscore. In both the AS and TCGA cohorts, only the AS-score and Immunoscore retained statistical significance in the multivariate models, underscoring their independent prognostic val-ue. Among the clinical variables, surgical margins (p=0.005) and tumor location (p=0.001) (in the AS cohort), as well as margins (p=0.04) and mitotic rate(p=0.04)  (in the TCGA cohort), showed significance in univariate analysis but did not remain significant in the multivariate context. These results suggest that the AS-score may provide additional, biology-driven prognostic insight be-yond conventional clinical features.”

  • Can the four-gene model (IGF1R, MAP2K1, SERPINE1, TCF12) be translated into a practical clinical test?

Response: Hypothetically, the four‑gene model offers a simple framework that could be readily translated into the clinical setting once validated. However, since RNA‑seq is not yet part of routine diagnostics, it will be essential to validate this signature using orthogonal, clinically feasible techniques—such as immunohistochemistry (IHC) or quantitative PCR (RT‑qPCR).

  • What are the clinical implications of identifying tumors with a high immunoscore and a high ASscore - are they suitable for therapeutic intervention?

Response: Based on our results, we hypothesize that patients with a high Immunoscore—indicative of robust immune infiltration—and a high AS‑score, reflecting poorer prognosis, may represent an especially promising group for immunotherapy. Their active immune‑infiltrated tumor microenvironment could sensitize them to immune checkpoint inhibitor therapy, and their unfavorable baseline prognosis suggests a potentially greater clinical benefit.

  • What is the mechanistic rationale behind the selection of genes for AS score - do they play a known role in immune evasion or tumor progression in sarcomas?

Response: The genes included in the AS‑score were selected based on their statistical association with overall survival, rather than on prior biological assumptions. However, the gene panel used in our RNA‑seq experiments comprises exclusively immune‑oncology–related genes. Thus, while the four selected genes were chosen empirically, they are all likely involved in immunomodulation. This approach, selecting prognostic genes from a pre-defined immune-related panel, is well aligned with similar prognostic models in oncology research. Many validated prognostic signatures stem from immune gene panels and retain biological relevance despite empirical selection.

Gene

Function

Role in Cancer (including Sarcomas)

Link to Immune Evasion / Tumor Progression

IGF1R

Receptor tyrosine kinase activating PI3K/AKT and MAPK pathways.

Overexpressed in several sarcomas (e.g., Ewing sarcoma, osteosarcoma). Involved in cell proliferation and survival.

Supports tumor growth and resistance to therapy. Shown to contribute to immune evasion by modulating tumor microenvironment.

MAP2K1

Kinase in the RAS/RAF/MEK/ERK signaling cascade.

Mutations and pathway activation reported in GISTs and other soft tissue sarcomas.

Promotes proliferation and may alter expression of immune-related genes, affecting tumor immunogenicity.

SERPINE1

Inhibitor of fibrinolysis; regulates ECM remodeling and angiogenesis.

Upregulated in soft tissue sarcomas; associated with poor prognosis and metastatic behavior.

Facilitates invasion and may contribute to immune suppression via extracellular matrix remodeling.

TCF12

bHLH transcription factor involved in cell differentiation and EMT.

Implicated in tumor progression and metastasis in multiple cancers. Limited but emerging evidence in sarcomas.

Associated with EMT and invasiveness, which may promote immune escape by altering cell phenotype and antigen presentation.

  • Is the observed correlation between AS score and immunoscore indicative of causality, or could it reflect overlapping but independent immune activity?

Response: While correlation does not imply causation, our multivariate Cox regression analyses, including both the AS‑score and Immunoscore as covariates, demonstrate that these two scores provide independent prognostic information. This finding supports the hypothesis that they reflect distinct, yet potentially complementary, biological processes.

  • Additional Figure 6: Multivariate Cox Regression Analysis Demonstrating the Independent Prognostic Value of AS-score and Immunoscore
  • Section 3.4. Combined Stratification by AS-Score and Immunoscore, line 255 “Furthermore, multivariate Cox regression analyses, incorporating both the AS-score and Im-munoscore as covariates, confirmed that each provides independent prognostic information. This supports the hypothesis that the AS-score and Immunoscore capture distinct, yet potentially com-plementary, biological processes relevant to tumor progression and immune contexture (Figure 6).”
  • How should the limitations of the model, particularly its retrospective design and single-region sample collection, affect its clinical application?

Response: While the retrospective and single-region design of our study may limit its immediate applicability in clinical settings, it serves as a valuable starting point for developing a clinically applicable model.

  • What ethical safeguards should be put in place before using the AS score as an inclusion criterion for clinical trials?

Response: Given the nature of the model, the ethical safeguards to be implemented align with those considered when evaluating any other prognostic or response-predictive biomarker in a clinical trial.

  • Could the AS score and immunoscore be adapted or retrained for other aggressive sarcoma subtypes?

  • Response: The AS‑score and Immunoscore may have utility in other aggressive sarcoma subtypes. Preliminary results from the TCGA sarcoma cohort suggest broader applicability. Nevertheless, we recommend that future studies validate both scores individually in each sarcoma subtype, including potential recalibration of the cut‑off thresholds according to disease context.

  • Section 4. Discussion, line 289: “… However, future studies should validate each score individually across different histological subtypes, with consideration for potential recalibration of classification thresholds to account for disease-specific biological contexts.”

Reviewer 3 Report

Comments and Suggestions for Authors

The authors sought to develop a gene expression–based prognostic model for angiosarcomas, termed the AS-score, and validate it in a broader sarcoma cohort using TCGA data. While the AS-score appears to stratify patients by outcome, its clinical utility is constrained by several limitations, including a small training cohort (n=25) and insufficient methodological clarity. Moreover, the reported correlation with the Immunoscore does not convincingly demonstrate added clinical value.

Comments below:

  • The authors should include the follow up time for all groups used for survival analysis. 
  • The authors mention that they have information about patient age, tumor size, medical history, anatomical location, treatment modalities (surgery, chemotherapy, radiotherapy), type of surgical procedure. This information should also be include in table 1
  • In the methods section, the authors state that the AS-score was based on the expression of candidate immune-related genes; however, they should provide more detail on how these candidates were selected. It is unclear from the citation or the study which specific genes were initially considered and on what basis they were identified as candidates.
  • In the Methods section describing the development and validation of the prognostic AS-score, the authors do not provide sufficient detail to ensure the reproducibility of their results. Clearer information on gene selection criteria, model construction, and statistical thresholds is needed to allow independent validation. More specifically:
    • The authors state that a multivariate Cox regression model was used to assign weights to the selected genes; however, key methodological details are missing. It is unclear which genes were included in the model, how the model was applied, what covariates (if any) were considered, and where the corresponding results can be found. 
    • The number of input genes used in the analysis is not defined and must be clearly stated.
    •  Furthermore, the authors should provide evidence—either as a table or graphical summary—showing the performance of all genes analyzed, including those filtered out and the 10 most significant genes ultimately used to train the Cox model. This information is essential for understanding and replicating their gene selection process.
  • Figure 3. What is the HR (Hazard Ratio) for the cohort? What about a multivariate model?
  • In section: 3.2. Validation of the AS-Score in the TCGA Sarcoma Cohort
    • What is the median follow-up time for the TCGA cohort? The authors need to provide more detailed information about the TCGA cohort, including clinical and demographic characteristics, follow-up duration, and outcome data. Additionally, a clearer comparison between the TCGA cohort, the authors' own cohort, and the cohort used for initial gene selection is necessary to contextualize the analysis and assess the applicability and generalizability of the AS-score.
  • Under section - Prognostic Relevance and relation to the Immunoscore
    • Is there any overlap between the genes included in the Immunoscore signature and those used in the AS-score? The authors should clarify this point and discuss its implications. Additionally, they should report the correlation between the two scores, including the R-squared value, to better characterize the relationship and assess whether the AS-score provides independent prognostic information beyond the Immunoscore.
    • What is the hazard ratio (HR) of a combined model that includes both the AS-score and the Immunoscore? The current data suggest that the two scores may be capturing similar information. The authors should demonstrate whether the AS-score offers added prognostic value beyond the Immunoscore alone, either by comparing model performance or by showing that the combined model significantly improves risk stratification.
  • Line 222: The authors should ensure consistency in referencing figures throughout the manuscript e.g. Graphic vs figure. Also the reference seems to be incorrect and probably referring to figure 4b

Author Response

Reviewer 3: (please also see the PDF attached)

Comments and Suggestions for Authors

The authors sought to develop a gene expression–based prognostic model for angiosarcomas, termed the AS-score, and validate it in a broader sarcoma cohort using TCGA data. While the AS-score appears to stratify patients by outcome, its clinical utility is constrained by several limitations, including a small training cohort (n=25) and insufficient methodological clarity. Moreover, the reported correlation with the Immunoscore does not convincingly demonstrate added clinical value.

We hereby submit our revised manuscript to be reconsidered for publication in IJMS following a revision in accordance with the reviewers’ comments.

We would like to thank the reviewers for the evaluation of our manuscript and for their constructive suggestions which we believe have contributed towards improving the article.

Please find below our response to the comments. The adapted sections of the article are highlighted in yellow in the manuscript.

Comments below:

  • The authors should include the follow up time for all groups used for survival analysis. 
    • Response: Thank you for your suggestion. We have now included the follow-up durations for all relevant subgroups in both the AS and TCGA cohorts, as detailed below.
  • AS score: TCGA median (range)
    • High (1): 363 (0-2324) days
    • Low (0): 591.5 (0-5324) days
  • IS score: TCGA
    • High (1): 704.5 (0-4056) days
    • Low (0) : 485 (0-5324) days
  • Combined score: TCGA
    • G1: 492 (0-5324) days
    • G2: 836 (0-4056) days
    • G3: 404 (0-2324) days
    • G4: 338.5 (22-1521) days

  • AS score: AS series median (range)
    • High (1): 10.09 (7.48-11.59) months
    • Low (0): 9.86 (0-11.56) months
  • The authors mention that they have information about patient age, tumor size, medical history, anatomical location, treatment modalities (surgery, chemotherapy, radiotherapy), type of surgical procedure. This information should also be include in table 1

Response: We appreciate this recommendation and have updated Table 1 to include key demographic and clinical variables such as age, tumor size, anatomical location, treatment modalities, and surgical procedures, where available.

  • In the methods section, the authors state that the AS-score was based on the expression of candidate immune-related genes; however, they should provide more detail on how these candidates were selected. It is unclear from the citation or the study which specific genes were initially considered and on what basis they were identified as candidates.

Response: The immune-related genes referenced in the Methods section correspond to the complete 1,392-gene panel included in the commercial HTG EdgeSeq Precision Immuno-Oncology (PIO) platform. No gene pre-selection occurred at this stage; instead, the entire panel was subjected to statistical filtering as described below.

  • In the Methods section describing the development and validation of the prognostic AS-score, the authors do not provide sufficient detail to ensure the reproducibility of their results. Clearer information on gene selection criteria, model construction, and statistical thresholds is needed to allow independent validation. More specifically:
    • Response: We appreciate the reviewer’s suggestion. Given that the steps to obtain the proposed model were not sufficiently clear, we will add detailed information addressing all the highlighted points to the appropriate sections of the Methods.
    • Section 2.3: Development and Validation of Prognostic AS-Score, Line 156 “The AS-score was derived from the expression data of candidate immune-related genes across 25 angiosarcoma (AS) cases [12]. A multi-step analytical approach was employed to construct the model. Initially, the prognostic significance of each of the 1,392 genes under consideration was assessed using the Maxstat algorithm (R package maxstat, v0.7-25), which determines the optimal cut-off based on log-rank test statistics for overall survival, thereby identifying genes with the strongest univariate associations with patient outcomes. Following this screening, the ten genes exhibiting the highest statistical significance, indicated by the lowest p-values, were selected for further analysis. These top ten candidates were subsequently entered into a multivariate Cox regression model to evaluate their independent prognostic value and to assign their respective weights. As a result of this analysis, four genes—IGF1R, MAP2K1, SERPINE1, and TCF12—were identified. Based on these four genes, the prognostic risk score, referred to as the AS-score, was developed. The AS-score was defined as the sum of the normalized expression values of the four genes, each weighted by its corresponding regression coefficient derived from the multivariate Cox analysis:

AS-score = (IGF1R × –1.22281) + (MAP2K1 × –0.63790) + (SERPINE1 × 1.00977) + (TCF12 × 0.61161)

A cut-off value of –1.9525 was established based on the threshold that most effectively stratified the population according to overall survival. This value was subsequently used to dichotomize patients into high- and low-risk groups.

  • Independently, trying to explain the followed steps to develop the model, the suggested questions will be answer one by one:
    • The authors state that a multivariate Cox regression model was used to assign weights to the selected genes; however, key methodological details are missing. It is unclear which genes were included in the model, how the model was applied, what covariates (if any) were considered, and where the corresponding results can be found. 

Response:

      • Maxstat Filtering: All 1,392 genes were evaluated using the Maxstat algorithm (R package maxstat, v0.7-25) to determine prognostic significance based on overall survival. This algorithm identifies the optimal cut-off point by maximizing the log-rank test statistic, facilitating the selection of genes with the strongest univariate associations with patient outcomes.
      • Top 10 Selection: The ten most statistically significant genes were selected based on their p-values for input into the Cox regression analysis.

Multivariate Cox Model: These ten genes were analyzed using multivariate Cox regression to evaluate their independent prognostic value and to assign their respective weights.

      • AS-Score Construction: The AS-score was defined as a linear combination of the expression levels of these four genes (IGF1R, MAP2K1, SERPINE1, TCF12), each weighted by its corresponding regression coefficient derived from the multivariate Cox analysis. AS-score = (IGF1R × –1.22281) + (MAP2K1 × –0.63790) + (SERPINE1 × 1.00977) + (TCF12 × 0.61161)

These details have been incorporated into the manuscript.

    • The number of input genes used in the analysis is not defined and must be clearly stated.

Response:

    • Step 1: Maxstat Filtering

All 1,392 genes were evaluated using the Maxstat algorithm (R package maxstat, v0.7-25) to determine prognostic significance based on overall survival. This algorithm identifies the optimal cut-off point by maximizing the log-rank statistic, facilitating the selection of genes with the strongest univariate associations with patient outcomes.

    • Step 2: Cox Regression Analysis

The ten most statistically significant genes, as determined in Step 1, were subjected to multivariate Cox regression analysis. This analysis assessed their independent prognostic value and assigned regression coefficients to each gene.

    • Step 3: AS-Score Construction
    • The AS-score was constructed as a linear combination of the expression levels of the four genes identified in Step 2 (IGF1R, MAP2K1, SERPINE1, and TCF12), each weighted by its corresponding regression coefficient derived from the Cox regression analysis.
    • Furthermore, the authors should provide evidence—either as a table or graphical summary—showing the performance of all genes analysed, including those filtered out and the 10 most significant genes ultimately used to train the Cox model. This information is essential for understanding and replicating their gene selection process.

Response: Due to the large size of the initial gene panel (1,392 genes), we chose to present the Maxstat and Cox regression results for the ten and four most significant genes, respectively. These results are now included in the supplementary tables. We trust that this information sufficiently supports the accuracy of the data presented in the manuscript.

Table S2: The 10 most significant genes evaluated by the maxstat algorithm.

Gene

Cut-off

p-value

OS p-value

SERPINE1

13.66

0.058

1.54*10-6

CD200

12.28

0.011

7.05*10-4

HES1

13.34

0.001

8.47*10-4

SMAD2

13.55

0.003

1.17*10-3

MT2A

16.27

0.212

1.30*10-3

CCND1

15.33

0.014

1.36*10-3

IGF1R

12.75

0.001

1.64*10-3

PDK2

11.55

0.019

1.70*10-3

TCF12

13.82

0.002

1.77*10-3

MAPK1

13.25

0.015

1.81*10-3

Table S3: Significant Gene-performance in cox regression analysis.

Gene

HR

p-value

weight

SERPINE1

2.32

<0.001

1.00977

IGF1R

0.32

0.001

–1.22281

TCF12

1.78

0.133

0.61161

MAPK1

0.44

0.026

–0.63790

  • Figure 3. What is the HR (Hazard Ratio) for the cohort? What about a multivariate model?

Response: The hazard ratio (7) and corresponding p-value (<0.001) for the AS cohort are presented in the following figure.

    • Response:In the multivariate model, the AS-score was constructed solely from gene expression data, without incorporating additional covariates.
  • In section: 3.2. Validation of the AS-Score in the TCGA Sarcoma Cohort
    • What is the median follow-up time for the TCGA cohort? The authors need to provide more detailed information about the TCGA cohort, including clinical and demographic characteristics, follow-up duration, and outcome data. Additionally, a clearer comparison between the TCGA cohort, the authors' own cohort, and the cohort used for initial gene selection is necessary to contextualize the analysis and assess the applicability and generalizability of the AS-score.

Response: The median follow-up time for the TCGA cohort (N=253) was 550 days (range: 0–5,324 days). The complete demographic information is included in Supplementary Table 1.

  • Under section - Prognostic Relevance and relation to the Immunoscore
    • Is there any overlap between the genes included in the Immunoscore signature and those used in the AS-score? The authors should clarify this point and discuss its implications. Additionally, they should report the correlation between the two scores, including the R-squared value, to better characterize the relationship and assess whether the AS-score provides independent prognostic information beyond the Immunoscore.

Response: There is no overlap between the genes included in the AS‑score and those used to calculate the Immunoscore. Each score serves a distinct purpose: the Immunoscore is designed to predict immune cell infiltration and act as a surrogate for immunotherapy response, whereas the AS‑score is trained to predict patient prognosis. However, as we propose in the paper, combining both scores may provide complementary insights into prognosis and treatment response, thereby improving the selection of candidates for immunotherapy. In line with the reviewer’s suggestion, we have now assessed the correlation between the two scores in the TCGA cohort. The results—including R² and p‑value—are shown in the following figure.

    • What is the hazard ratio (HR) of a combined model that includes both the AS-score and the Immunoscore? The current data suggest that the two scores may be capturing similar information. The authors should demonstrate whether the AS-score offers added prognostic value beyond the Immunoscore alone, either by comparing model performance or by showing that the combined model significantly improves risk stratification.

Response: To assess whether the AS‑score adds prognostic value beyond the Immunoscore, we performed a multivariate Cox regression analysis including both scores as covariates. The results demonstrate that each score retains independent prognostic significance. These findings are illustrated in the following figure.

  • Additional Figure 6: Multivariate Cox Regression Analysis Demonstrating the Independent Prognostic Value of AS-score and Immunoscore
  • Section 3.4. Combined Stratification by AS-Score and Immunoscore, line 255 “Furthermore, multivariate Cox regression analyses, incorporating both the AS-score and Im-munoscore as covariates, confirmed that each provides independent prognostic information. This supports the hypothesis that the AS-score and Immunoscore capture distinct, yet potentially com-plementary, biological processes relevant to tumor progression and immune contexture (Figure 6).”

We can now confirm that both scores function independently and provide distinct prognostic information.

  • Line 222: The authors should ensure consistency in referencing figures throughout the manuscript e.g. Graphic vs figure. Also, the reference seems to be incorrect and probably referring to figure 4b.

Response: We thank the reviewer for noting this inconsistency. The reference to “Graphic 3B” has been corrected to “Figure 3B” in the revised manuscript, and we have reviewed the entire text to ensure consistency in figure citations.

Round 2

Reviewer 1 Report

Comments and Suggestions for Authors

The authors should probably cite some references on deep learning in their revision since this would be helpful to the general reader.

Author Response

Response: References related to machine learning have been added

Reviewer 2 Report

Comments and Suggestions for Authors

The authors responded to my concerns.

Author Response

Response: Thank you for the comments

Reviewer 3 Report

Comments and Suggestions for Authors
  • Figure 3: The authors should adjust for additional demographic and clinical variables—such as age, gender, primary tumor site, tumor size, stage, surgical resection, and treatment modalities—to more robustly demonstrate the independent prognostic value of the AS score.
  • Title: The title has a typo in the word immunoscore 
  • In Table S2,The authors present the top 10 most significant genes, yet MT2A does not appear to be statistically significant. Could the authors clarify why it was included in the list?
  • Similarly, in Table S3, TCF12 does not demonstrate significant performance, yet it is included in the table and selected as part of the prognostic model. Could the authors clarify the rationale for its inclusion?
  • In Figure 4 and the related analyses, the authors state that the AS score is an independent prognostic tool; however, this claim is not fully supported by the results presented. To demonstrate true prognostic independence, the AS score should retain significance in a multivariate analysis that includes key clinical covariates such as age, gender, primary site, tumor size, tumor stage, surgical intervention, and therapies. The authors should provide hazard ratios and p-values from such a model to substantiate this claim.

Author Response

  1. Figure 3:The authors should adjust for additional demographic and clinical variables—such as age, gender, primary tumor site, tumor size, stage, surgical resection, and treatment modalities—to more robustly demonstrate the independent prognostic value of the AS score.
  • Response: We thank the reviewer for this helpful suggestion. To more rigorously assess the independent prognostic value of the AS score in our angiosarcoma cohort, we conducted univariate and multivariate Cox proportional hazards analyses, incorporating the following clinical variables:
  • Age
  • Gender
  • Primary tumor site
  • Tumor size
  • Mitotic rate
  • Surgical resection
  • Chemotherapy
  • Resection margins

The results, summarized in the following table, indicate that the AS score remained statistically significant in the multivariate model (HR = 7.0; 95% CI: 2.4–21; p = 0.000444), whereas the other clinical variables did not show independent significance. These findings support the AS score as an independent prognostic factor in our angiosarcoma cohort.The table has been added to the supplementary materials as Supplementary Table 4. Additionally, the results have been incorporated into Section 3.1, Development of the AS Score in the AS Series, line 216, as follows:

  • “To further validate the prognostic value of the AS score, we conducted both univariate and multivariate Cox regression analyses, incorporating common clinicopathological parameters alongside the AS score (Supplementary Table 4). In the univariate analysis, primary tumor site (cutaneous vs. non-cutaneous; p = 0.0012), surgical margin status (p = 0.0049), and chemotherapy treatment (p = 0.0003) were significantly associated with over-all survival (OS). However, in the multivariate model, only the AS score remained an inde-pendent predictor of poor prognosis (HR = 7.0; 95% CI: 2.4–21; p = 0.000444), while the other clinical variables lost statistical significance. These findings support the robustness of the AS score as a prognostic biomarker that provides independent predictive value be-yond conventional clinicopathological features.”
  1. Title: The title has a typo in the word immunoscore 

Response: The typo has been corrected and the title has been updated.

  1. In Table S2, the authors present the top 10 most significant genes, yet MT2A does not appear to be statistically significant. Could the authors clarify why it was included in the list?

Response: Although MT2A did not reach statistical significance by the Maxstat algorithm, it was included due to its prognostic relevance identified via univariate survival analysis using the log‑rank test—our primary gene selection criterion.The Maxstat algorithm was specifically used to determine the optimal expression cutoff for each gene, selecting the threshold that best separated the patient cohort into two groups with significantly different overall survival.For MT2A, after establishing that cutoff with Maxstat, the resulting dichotomized groups demonstrated a statistically significant survival difference by log‑rank analysis, thereby justifying its inclusion among the top‑ranked genes.

  1. Similarly, in Table S3, TCF12 does not demonstrate significant performance, yet it is included in the table and selected as part of the prognostic model. Could the authors clarify the rationale for its inclusion?

Response: We appreciate the reviewer’s comment regarding the inclusion of TCF12 in the prognostic model despite its lack of individual statistical significance. To clarify, the initial selection of the 10 genes—including TCF12—was based on their significant association with survival in univariate Cox regression analysis. These 10 genes were subsequently included in a multivariable Cox model to evaluate their combined prognostic value. In the multivariable model, four genes were retained, including TCF12, which remained in the final model despite not reaching individual statistical significance. Although TCF12 did not achieve conventional statistical significance (p > 0.05), it was retained based on overall model performance criteria, including the Akaike Information Criterion (AIC) and log-likelihood, used during the variable selection process. Thus, even though TCF12 was not statistically significant on its own, its inclusion enhanced the stability and predictive performance of the final model. Moreover, it is well established that variables with borderline or non-significant p-values can still contribute meaningful information when considered jointly with other predictors—especially in complex biological systems such as cancer. For construction of the prognostic score, we used the regression coefficients from the final multivariable model, as this approach reflects the combined effect of all retained variables and more accurately captures the multivariate relationships among them.

  1. In Figure 4 and the related analyses, the authors state that the AS score is an independent prognostic tool; however, this claim is not fully supported by the results presented. To demonstrate true prognostic independence, the AS score should retain significance in a multivariate analysis that includes key clinical covariates such as age, gender, primary site, tumor size, tumor stage, surgical intervention, and therapies. The authors should provide hazard ratios and p-values from such a model to substantiate this claim.

Response: To demonstrate the independent prognostic value of the AS score in the TCGA sarcoma dataset, we performed univariate and multivariate Cox regression analyses, including the following covariates:

Age; Gender; Tumor size; Mitotic rate;, Radiotherapy and Resection margins

As summarized in the table below, the AS score remained statistically significant in the multivariate model (HR = 2.47; 95% CI: 1.42–4.30; p = 0.00142), alongside the Immunoscore—confirming its prognostic value independent of standard clinicopathological variables. A table detailing these results has been added to the supplementary materials as Supplementary Table 5. Additionally, we have updated the manuscript in Section 3.2, “Validation of the AS‑Score in the TCGA Sarcoma Cohort” (line 276), to reflect these findings.

  • To rigorously assess whether the AS‑score offers prognostic value independent of estab-lished clinicopathological variables, we conducted an extended multivariate Cox regres-sion analysis in the TCGA sarcoma cohort, adjusting for age, gender, tumor size, mitotic rate, radiotherapy, and surgical margin status (Supplementary Table 5). Significantly, both the AS‑score (HR = 2.47; 95% CI: 1.42–4.30; p = 0.0014) and the Immunoscore (HR = 0.42; 95% CI: 0.25–0.70; p = 0.0009) retained statistical significance in this comprehensive mod-el, whereas none of the clinical variables, including tumor size, mitotic rate, or surgical margin status did. These findings confirm that the AS‑score is an independent prognostic marker, offering robust, biology-driven prognostic information that complements tradi-tional clinical features in soft tissue sarcomas.

Round 3

Reviewer 3 Report

Comments and Suggestions for Authors
  • In Supplemental table 5 there is typo in the column names univariate and multivariate analysis.